# Blastocyst-Derived Lactate as a Key Facilitator of Implantation

**DOI:** 10.3390/biom15010100

**Published:** 2025-01-10

**Authors:** Kathryn H. Gurner, David K. Gardner

**Affiliations:** 1Melbourne IVF, East Melbourne, VIC 3002, Australia; kathryn.gurner@mivf.com.au; 2School of Biosciences, University of Melbourne, Parkville, VIC 3010, Australia

**Keywords:** embryo, endometrial receptivity, IVF, metabolism, metaboloepigenetics

## Abstract

The blastocyst develops a unique metabolism that facilitates the creation of a specialized microenvironment at the site of implantation characterized by high levels of lactate and reduced pH. While historically perceived as a metabolic waste product, lactate serves as a signaling molecule which facilitates the invasion of surrounding tissues by cancers and promotes blood vessel formation during wound healing. However, the role of lactate in reproduction, particularly at the implantation site, is still being considered. Here, we detail the biological significance of the microenvironment created by the blastocyst at implantation, exploring the origin and significance of blastocyst-derived lactate, its functional role at the implantation site and how understanding this mediator of the maternal–fetal dialogue may help to improve implantation in assisted reproduction.

## 1. Embryo–Endometrial Dialogue Facilitates Implantation

The implantation of the blastocyst into the maternal endometrium is critical in the establishment of a successful pregnancy. The success of this elegant biological phenomenon requires both the embryo and endometrium to be appropriately developed and to have established a reciprocal dialogue, mediated by a number of soluble factors present within the uterine microenvironment. Factors including glycosaminoglycans, miRNAs and extracellular vesicles, enzymes, hormones, growth factors (GFs) and cytokines originating from both the endometrial glandular epithelium and stoma, as well as from the blastocyst itself, act through both paracrine and autocrine mechanisms to invoke phenotypic or functional changes in both cell types. Collectively, these changes ensure that the endometrium is in a period of receptivity at a time when a viable blastocyst is present, to facilitate successful implantation. Recently, the role of metabolites as signaling molecules has gained attention, whereby specific metabolites can promote cellular functional changes, similar to classical signaling factors. However, the role of these metabolites in the embryo–endometrial dialogue remains largely unexplored. Therefore, here we consider how one such metabolite, lactate, plays multifactorial roles in facilitating communication between the embryo and the endometrium to support implantation.

## 2. Embryonic Secretome: Signaling for Implantation

Key studies on the maternal (endometrial) side of this dialogue have concentrated on the role of classical signaling molecules (e.g., growth factors and cytokines) in synchronizing the development of the embryo and the endometrium for a successful pregnancy. These have been extensively considered previously [1,2], and therefore further discussion is beyond the scope of this review. Recently, however, attention has shifted to the embryonic secretome which encompasses a wide array of small proteins, metabolites and other soluble factors which have the potential to participate in the embryo–endometrial dialogue at the implantation site [3]. These embryo-secreted factors have primarily been identified through analyses of spent embryo culture media and include proteases [4], miRNAs [5], platelet activating factor [6,7], human chorionic gonadotropin [8,9,10,11,12] and several other secretory proteins [3]. Many function to modify the adhesive capacity of endometrial epithelial cells and can alter the expression of specific genes and proteins that are associated with implantation [2,13,14].

In addition to more “traditional” signaling molecules (i.e., GFs and cytokines), the embryonic secretome also includes metabolites, such as amino acids, acetyl CoA, succinate and lactate (and its protonated form, lactic acid). While many of these molecules do not yet have defined roles at the implantation site, it is becoming increasingly recognized that certain metabolites, in addition to their roles in energy-generating pathways and as biosynthetic precursors, can also act as intracellular signaling molecules to alter the activity of specific pathways, or modulate transcription factor activity and promote cytokine production [15,16]. Additionally, it has been identified that metabolites, such as succinate [17], acetoacetate, ßOHB [18] and lactate [19,20], can act as extracellular ligands to activate specific G protein-coupled receptors (GPCRs), thus supporting the concept of metabolites as signaling molecules.

Metabolites can also influence epigenetic regulation, inducing long-term changes to the epigenome, in a process referred to as ‘metaboloepigenetics’ [21,22]. For example, key energy metabolites and cofactors including acetyl CoA [23,24], NAD^+^ [25,26] and alpha ketoglutarate [27,28] can serve as essential cofactors for epigenetic-modifying enzymes including histone deacetylases and acetyltransferases and DNA methyltransferases, while lactate, in addition to being able to regulate NAD^+^ levels, can induce a novel post-translational protein modification called lysine lactylation (Kla). This was first identified as an epigenetic mark, whereby lactyl groups are deposited on the lysine tails of histone proteins, increasing DNA accessibility and gene transcription [29]. Recent studies have shown that non-histone proteins within the nucleus, cytoplasm and mitochondria can also be lactylated, with downstream impacts on the regulation of numerous biological processes [30,31,32]. Significantly, it was also recently reported that metabolites are retained within chromatin following epigenetic modification, suggesting that epigenetic sites not only serve as modifications for gene regulation but also act as metabolite reservoirs. The retention of metabolites following acetylation was proposed to enable the rapid reallocation of metabolites, such as acetate groups, from one histone lysine site to an active transcription site according to cellular needs [33,34]. Additionally, this process helped replenish the acetyl CoA pool for use in metabolic pathways under conditions of low nutrient availability [35]. While not yet investigated, this could also be the case for lactate following histone lactylation. Thus, plausibly, embryo-derived metabolites may participate in endometrial–embryo communication to impact cell function and gene expression.

## 3. Lactate: Unmasking the Roles of a Neglected Metabolite

The mammalian blastocyst establishes an idiosyncratic metabolism in which approximately half of the glucose consumed by the blastocyst is converted to lactate, despite the blastocyst’s high oxidative capacity [36,37]. This metabolic trait, herein referred to as blastocyst-specific aerobic glycolysis, was initially reported in tumor cells by Otto Warburg in 1927 [38,39] and hence is also referred to as the “Warburg effect”. Warburg considered this lactate production in cancers to be an indication of mitochondrial dysfunction. However, this did not seem to be the case, as functioning mitochondria were subsequently identified in many cancers, indicating that the lactate produced by cancers may have a specific role. Research over the last 30 years has helped us understand that this metabolite has multiple biological functions beyond its conventional role in facilitating glycolytic flux, roles involved in promoting a variety of cellular changes and functions.

### 3.1. Metabolic Roles

Lactate is formed through a near-equilibrium reaction facilitated by the enzyme lactate dehydrogenase (LDH), in which pyruvate is consumed with the concomitant oxidation of NADH to NAD^+^ (Figure 1). The Gibb’s free energy of this reaction is close to zero, and therefore a major determinant as to the direction of flux through LDH is the relative concentration of pyrvuate:lactate. The regeneration of NAD^+^ facilitates continuous glucose flux through glycolysis where it is required as a cofactor by the enzyme glyceraldehyde-3-phosphate dehydrogenase, thereby supporting energy production, with any lactate formed typically being released into the extracellular environment. This maintains low intracellular concentrations of lactate allowing glycolysis to continue by mass action. In the blastocyst, lactate efflux is mediated by monocarboxylate transporters (MCTs) present on the apical surface of trophectoderm cells. Specifically, MCT1 [40] and MCT4 [41] are found on mouse blastocysts, while MCT1 and MCT2 are present on human blastocysts [42]. Since MCTs are proton-linked transporters, this process involves the co-transport of lactate (anion) and a proton (H^+^), resulting in the release of “lactic acid” into the extracellular environment. Historically, lactate has therefore been perceived as an end (or waste) product of glucose metabolism. This is exemplified in typical biochemistry texts, which typically overlook the roles of this simple three-carbon molecule, focusing much more on the importance of glucose and pyruvate in metabolism.

In addition to the provision of ATP for the energetically demanding processes of blastocoel expansion and cellular proliferation/development, high levels of glucose flux and the subsequent production of lactate act to ensure that there is sufficient glucose carbon to enter the pentose phosphate pathway (PPP) [43] (Figure 1). This in turn facilitates the creation of biosynthetic precursors such as triacylglycerols, phospholipids and ribose moieties for lipid and nucleic acid biosynthesis [43,44]. The generation of NADPH through this pathway is also vital for fatty acid and sterol synthesis [44], and for the production of reduced glutathione, an intracellular antioxidant that supports pre-implantation embryonic development [45].

Furthermore, lactate generated through glycolysis can also act as an allosteric regulator of phosphofructokinase (PFK), a rate-limiting enzyme in glycolysis that catalyzes the conversion of fructose-6-phosphate to fructose-1,6-bisphosphate (Figure 1). Through promoting the dissociation of tetrameric PFK into inactive dimeric PFK [46], lactate can decrease the catalytic activity of PFK [46,47,48] and thus regulate glycolytic flux.

As well as facilitating metabolic requirements for rapid proliferation, high levels of blastocyst-specific aerobic glycolysis have also been interpreted to represent an embryonic adaptation that enables the embryo to proceed through the initial phases of implantation [43,49]. Microvasculature studies in rodents have shown that the initial stages of implantation occur under anoxic conditions, due to the lack of a vascular bed in the immediate vicinity of the implantation site [50,51]. Therefore, at the time of implantation itself, the levels of oxygen available to the embryo as it invades the endometrium are limited, resulting in a need to transition to a fully anerobic metabolism for the generation of energy. Hence, at the blastocyst stage, the utilization of blastocyst-specific aerobic glycolysis, together with the expression of glycolytic isoenzymes required for anaerobic glycolysis [52,53], aids in preparing the blastocyst to function metabolically in an anoxic environment.

### 3.2. Creation of a Specialized Implantation Microenvironment to Facilitate Signaling

Viable human blastocysts consume ~150 pmol of glucose per hour [54,55], which is approximately 10-fold higher than that of the cleavage-stage embryo [56,57]. Consequently, in the small volume of endometrial fluid surrounding the embryo, the blastocyst has the capacity to create a high concentration of lactate, thereby creating a microenvironment characterized by high lactate levels and low pH [36,57,58]. Accumulation of lactate and H^+^ in the microenvironment of tumor cells, which, as discussed, share a similar metabolism with the blastocyst, has been shown to play an important role in the promotion of cellular invasion and migration, signaling for angiogenesis and immunosuppression [59,60,61], cellular processes that are vital for implantation success. In fact, it has been postulated that cancers have adopted the elegant mechanisms employed by the blastocyst at implantation to prolong their proliferative capacity and ultimate survival [62,63,64,65,66]. Notably, in vitro studies utilizing endometrial cell lines have demonstrated a role for this specialized environment of high lactate and low pH in promoting functional changes to endometrial epithelial, stromal and endothelial cells to facilitate implantation [67]. The treatment of endometrial epithelial cells with lactic acid promotes the acquisition of endometrial receptivity by decreasing cellular proliferation and tight-junction integrity, while also upregulating the mRNA expression of endometrial receptivity markers *VEGF* and *SNAI1* [67]. Interestingly, this effect was lost following treatment with altered pH or neutralized lactic acid, implying that both high lactate and lowered pH in the microenvironment at the implantation site are important in facilitating functional changes for endometrial receptivity. While further research is required, these works collectively highlight the potential of lactic acid as an important mediator in the maternal–fetal dialogue to promote implantation.

## 4. Lactic Acid: Dual Functionality of Lactate and Its Associated Proton

Lactate and/or lactic acid (LA) can induce changes to cellular functions known to be important for endometrial remodeling required for implantation in both an indirect and direct manner (Figure 2). The terms lactate and LA are often used interchangeably in the literature; however, distinction between the two entities is important as they exert different biological effects, acting both as a substrate and as a proton donor. Moreover, due to its low pKA value (acid dissociation constant: ~3.86), under conditions of physiological pH, LA dissociates into the lactate anion and proton (H^+^). As such, there are actually three molecules to consider: lactate (anion), H^+^ (proton) and the fully formed acid (LA), when discussing its functional roles. In the context of this review, reference to lactate will hereafter refer to the lactate anion, while specific reference to acidification of the extracellular milieu will be used to discuss the dual functionality of lactate and its associated proton in the facilitation of several novel functions.

### 4.1. Lactate as an Extracellular Ligand

Lactate can function as an extracellular agonist/ligand for the G protein-coupled receptor GPR81 [68] (Figure 2A). Initially classified as an ‘orphan’ GPCR, work conducted by Liu and colleagues in 2009 identified lactate as the endogenous ligand for GPR81, capable of activating GPR81 at concentrations of 0.1–30 mM [20,69]. Subsequently, GPR81 has been shown to be expressed and active in several cell types and tissues including adipocytes [19,20], skeletal muscle [70], brain [71], liver [72] and a number of cancer cell lines [73] in which lactate has also been demonstrated to exert functional and signaling effects. GPR81 is coupled to the Gi signaling pathway [74], and, upon activation, can reduce cAMP levels and attenuate Akt signaling to promote cell survival and growth [19,20,75] (Figure 2A). Additionally, in breast cancer studies, activation of the β subunit promotes PI3K/AKT signaling and transcription of pro-angiogenic factors [76]. GPR81 activation can also suppress inflammatory cytokine activation in the liver, colon and myometrium [72,77,78], with implications for immunosuppression and cell survival, and can induce the expression of genes involved in lactate production, transport (e.g., MCTs, GLUTs) and metabolism (e.g., LDH) [73,79,80].

Current knowledge regarding the presence of GPR81 and the potential role of lactate signaling events within the reproductive tract is limited. Of significance, however, is that in human placentas, GPR81 has been identified on the basal side of the cytotrophoblast, suggesting that GPR81 and lactate may be involved in syncitialization [81] by modulating cAMP availability, an important activator of this process [82]. Further, in mice, increased lactate production during labor activates GPR81 within the myometrium, downregulating certain proinflammatory genes (e.g., *Il1b*, *Il6*, *Ccl2* and *Pghs2*). This effect is lost with the administration of the GPR81 agonist 3,5-dihdroxygenzoic acid [77]. Further studies are necessary, however, to investigate the presence of GPR81 and its potential interactions with lactate within the oviduct and uterus and establish the importance of this signaling paradigm in early pregnancy.

Of note, lactate can also activate another GPCR known as GPR132 or G2A [83,84,85]. Investigations into lactate signaling via GPR132 in human cells are quite limited. However, it has been shown in breast cancer cells that the activation of GPR132 in tumor-associated macrophages promotes their polarization to the non-inflammatory M2-like phenotype, thereby enhancing tumor growth [83,86]. Notably, GPR132 is recognized as a member of the “pH-sensing” GPCRs known to be responsive to changes in pH, resulting in their activation by the protonation of multiple histidine residues [85]. Thus, it remains unclear whether lactate activates GPR132 signaling through direct interaction with GPR132 or because of extracellular acidosis induced by lactate.

### 4.2. Lactylation: Post-Translation Protein Modification

Lactate can directly influence gene expression through a post-translational modification (PTM) known as lactylation (Kla) (Figure 2B). This was first discovered by Zhang and colleagues in 2019, who demonstrated that lactyl groups can be added to the lysine tails of histone proteins in MCF-7 cancer cells [29]. They found that the level of histone lactylation at specific sites, such as H4K12la, is correlated with intracellular lactate concentration, with the inhibition of lactate production (utilizing 2-deoxy-glucose) reducing the presence of these marks. Lactylation can also occur on non-histone proteins within the nucleus, cytoplasm and mitochondria, affecting various biological processes [30,31,32]. Notably, many of these proteins are enzymes involved in glycolysis and ribosomal functions [31]. Therefore, lactylation has now been confirmed as a prevalent PTM in numerous cellular contexts and tissues, with new histone and non-histone sites still being uncovered.

The role of lactylation in reproductive biology, however, is still somewhat unresolved. Histone lactylation during mouse oocyte maturation and embryonic development is temporally dynamic, with reports indicating that lactylation of various histone sites (including H3K23la, H3K18la, H4K15la, H4K12la and H3K14la) display distinct patterns across different embryonic developmental stages [87,88]. Interestingly, H3K18la is enriched in the promotor regions of a number of major zygotic genome activation (ZGA) genes in both mice and humans [89,90]. Accordingly, the loss of H3K18la marks during embryonic development, through the depletion of extracellular lactate, is linked with ZGA failure and developmental arrest at the two-cell stage. Notably, these effects were reversed with injection of lactyl-CoA, a substrate for histone lactylation [29], into zygotes deprived of lactate for 24 h, resulting in the subsequent reinstatement of H3K18la marks [90].

Furthermore, in the mouse uterus, lactylation is primarily localized to the myometrium and endometrium, with the highest levels reported in the endometrium [88], suggesting a possible role in endometrial function and implantation. This hypothesis has been explored further in a recent study by Zhao and colleagues, who showed that increased glycolysis and lactate production in mouse and human endometrial cells is linked to histone lactylation, particularly at H4K12la, with downstream impacts on endometrial stromal cell decidualization [91], a process that is essential for the establishment of a successful pregnancy [92]. They suggest that this occurs due to the existence of a positive feedback loop whereby H4K12la enhances the expression of HIF1α, a key transcription factor in decidualization. Further, they demonstrated that, in mice, the inhibition of lactate production and subsequent histone lactylation in the uterus through shRNA knockdown of *Ldha* prevented this effect, leading to increased resorptions and reduced post-transfer fetal weight and development [91]. Notably, this shRNA knockdown did not impact the expression of *Ldha* in mouse embryos [91] or their implantation potential; therefore, these findings support a positive role for lactylation in pregnancy establishment. Additionally, in neural crest cells, the lactylation of tissue-specific active promoters and enhancers [93,94] promotes the establishment of the neural crest gene regulatory network, subsequently promoting cellular differentiation, epithelial–mesenchymal transition (EMT) and cellular migration [94], functional changes that are also required for the acquisition of endometrial receptivity [95,96]. Therefore, it is possible that lactylation within other areas of the endometrium may also be important in mediating endometrial cellular function for the acquisition of endometrial receptivity.

### 4.3. Acidic pH and Influences on Cellular/Trophoblast Invasion

Environments of high lactate and/or high proton concentration (i.e., decreased external pH) are known regulators of the controlled degradation of extracellular matrix (ECM) proteins required for cellular invasion (Figure 2C). This is exemplified within the tumor microenvironment, where low external pH (pH ~6.7) is correlated with increases in tumor metastatic potential and cell survival [97,98,99]. This is of significance to the implanting blastocyst as the degradation of the surrounding endometrial ECM and subsequent invasion of the trophoblast towards the underlying endometrial stromal compartment is a key step in the establishment of implantation and pregnancy. The invasiveness of trophoblast cells is associated with the production of proteases, in particular MMPs [100], plasminogen activators [101] and cathepsins [102], which degrade ECM components and thus facilitate trophoblast penetration into the decidua. Notably, extracellular acidity can influence the expression and optimal function of such proteolytic factors. For example, the expression of MMP9 [103,104,105] and Cathepsin B [106] are increased by low external pH (pH of 5.4–6.5) in malignant melanoma cells in both mice [104] and human cell line [105] studies. These factors may, therefore, be responsive to lowered pH generated by the blastocyst at the implantation site. The proteolytic cascade that converts pro-cathepsins B and L [107,108] to their active states is also enhanced in environments of low pH, thus facilitating the efficient digestion of extracellular matrices. Further, in nucleus polposus cells and osteoarthritic studies, low external pH has been shown to downregulate the expression of TIMPs [109,110], thereby reducing the inhibition of MMP activity and increasing ECM breakdown/degradation. As such, the acidic microenvironment at the peri-implantation site, created by the efflux of blastocyst-derived LA, plausibly facilitates the activation and progression of proteolytic ECM digestion and resultant trophoblast invasion.

### 4.4. Lactate as an Intracellular Signaling Mediator: Promoting Angiogenesis in Wounds and Tumors

For implantation to be successful and pregnancy to be established, it is critical that a new blood supply is established following the invasion of the blastocyst through the endometrial luminal epithelium. This blood supply is vital for providing nutrients to the developing fetus prior to the formation of the placenta. In areas of tissue damage and within the tumor microenvironment, it is well documented that high levels of lactate accumulation play an important role in promoting angiogenesis and wound healing [111]. Functionally, this occurs through the uptake of extracellular lactate by surrounding cells, such as endothelial cells, macrophages and tumor support cells, via the action of MCTs. Once inside these cells, intracellular lactate can be converted to pyruvate, which indirectly stimulates angiogenesis through the activation and recruitment of major pro-angiogenic factors, such as vascular endothelial growth factor (VEGF) and the upregulation of pro-angiogenic signaling [112,113].

VEGF typically exists within cells as a mixture of both ADP-ribosylated (inactive, non-angiogenic) and free forms, the latter being primarily responsible for the documented pro-angiogenic actions of VEGF [114]. Notably, NAD^+^, which is significantly depleted during lactate oxidation (Figure 1), serves as an important substrate for ADP ribosylation, a process which involves the transfer of an ADP-ribose moiety from NAD^+^ to specific amino acid residues on certain proteins, such as arginine and glutamate [115,116,117]. Consequently, the conversion of lactate to pyruvate and the concomitant decrease in intracellular NAD^+^ can reduce the relative level of ADP-ribosylated VEGF within the cell, an action which has downstream pro-angiogenic effects (Figure 2D) [118,119].

Additionally, in tumors and endothelial cells, the MCT-driven import of lactate from the extracellular space and the resulting regeneration of intracellular pyruvate acts to inhibit the activity of prolyl hydroxylases (PHDs) (Figure 2D). This occurs through competitive inhibition, whereby pyruvate competes with the PHD co-substrate 2-oxoglutarate (α-ketoglutarate) for direct binding to PHD, thus inhibiting its activity. The functional inhibition of PHD promotes pro-angiogenic signaling through two primary mechanisms. Firstly, it prevents the activation of the PHD-mediated ubiquitin–proteasome pathway that normally targets hypoxia-inducible factors for degradation, and, as such, causes the stabilization of HIF1α [112,120,121]. HIF1α is a transcription factor that modulates the expression of ‘hypoxia responsive’ genes including *Glut1* and *Glut3* among other glycolytic enzymes [122,123], as well as *Igf2* and *Vegf*, that are involved in multiple cellular signaling cascades, including the promotion of angiogenesis and vascularization [124]. The stabilization of HIF1α in endothelial cells is also associated with increases in the protein expression of transmembrane VEGFR2 [112], the main transducer of VEGF pro-angiogenic effects. Therefore, lactate produced from the implanting blastocyst may act to upregulate VEGF production in endometrial endothelial cells through the stabilization of HIF isoforms to support the induction of angiogenesis. Secondly, the lactate-driven inhibition of PHD in endothelial cells also results in the accumulation of IkB kinase [125] and stimulates the release of active NFκB [113] (Figure 2D). This promotes the transcription of a number of genes, including pro-angiogenic IL8, which promote endothelial cell migration and subsequent vessel formation. This has been validated in vivo in mice, where injection of either MCT-silenced tumor cells [113] or PHD-silenced tumor cells resulted in a decrease in tumor growth and vessel formation in comparison to control tumors [126]. As such, indirectly, lactate signals to upregulate key genes and transcription factors which promote angiogenesis.

### 4.5. Lactate as an Immunomodulator

Lactate can impact the function of numerous immune cells known to be present in the decidua that collectively act to promote an immune-suppressive environment for the implanting embryo (Figure 2E). This includes members of the innate immune system family such as uterine natural killer cells [127], antigen presenting macrophages [128] and dendritic cells [129], as well as T reg cells of the adaptive immune system [130]. For instance, tumor-derived lactate has a well-defined role in promoting the functional polarization of macrophages from an inflammatory M1 phenotype to an immunosuppressive M2 phenotype through several mechanisms. This occurs following the MCT-driven uptake of lactate, where intracellular lactate can induce stabilization of HIF1α (as described above) to stimulate the expression of *Vegf* and *Arg1* and promote tumor cell growth [131,132]. The expression of anti-inflammatory markers (i.e., *Arg1*, *Fizz1* and *Vegf*) in tumor-associated macrophages can also be facilitated through the activation of mammalian target of rapamycin complex (MTORC) by intracellular lactate and the subsequent stabilization of HIF2α [133], or through the activation of ERK/STAT3 signaling [134]. Lactate within the tumor microenvironment can also be sensed by GPR132 present on the cell membrane of tumor-associated macrophages [83], leading to induction of cAMP signaling and the downstream promotion of pro-angiogenic macrophage phenotypes and associated gene transcription program/expression. More recently, histone lactylation has been identified as another mechanism by which lactate promotes M2 polarization and tumor growth by regulating homeostatic gene expression [29,135]. Hence, lactate produced from the implanting blastocyst can have a profound effect on the macrophage population phenotype in the endometrial decidua to assist in the establishment of ongoing pregnancy.

The accumulation of lactate in the tumor microenvironment, and subsequent decreases in external pH, has also been shown to mute the activity of inflammatory immune cells, and, thus, may play a role in enhancing the immune-suppressive population of cells within the uterine/decidual microenvironment. In cytotoxic T cells, decreased extracellular pH inhibits the production of proinflammatory cytokines from cytotoxic T cells (CD8+) and decreases cytotoxic activity by ~50%, an effect that can be reversed through the buffering of cellular pH [136,137]. Similarly, natural killer (NK) cells display significantly reduced cytotoxic activity following exposure to lactate or low pH [138,139], while the differentiation of monocytes to activated dendritic cells is also reduced in environments of low pH [140]. Further, lactate itself can induce TGFβ expression in glioma cells [141,142], which subsequently promotes T reg induction and differentiation [143], and, therefore, promotion of the immunosuppressive response. Consequently, blastocyst-derived lactate and parallel decreases in external pH appear to be important signals during early pregnancy to modulate decidual immune cells and facilitate immunotolerance.

## 5. Conclusions

Lactate, once perceived as a by-product of glycolysis, is now gaining increasing attention for its vital roles in promoting a variety of cellular changes and functions. This three-carbon carboxylic acid appears to be a metabolite of multiple functions. It is produced as an early signal from the developing blastocyst, transported rapidly across the trophectoderm cell membrane with a proton by MCTs and then readily enters surrounding tissues to alter their functions. Not only this, but lactate is also an established ligand for GPR81 and, further, can regulate gene expression through both histone and non-histone lactylation.

Given the discussed significance of lactate, it would be prudent to enhance our understanding of the novel roles that lactate and extracellular acidosis have at the maternal–fetal interface. Such information could not only help improve reproductive success, but could contribute to our understanding of cancer biology. For example, the role of GPR81–lactate interactions and signaling within both the endometrium and other reproductive tissues could be analyzed through the use of GPR81 agonists, such as AZ′5538 [144], or an siRNA knockdown in endometrial cells combined with functionality assays. Additionally, analysis of lactylation in the human endometrium could provide insights into the regulation of endometrial receptivity.

Hence, lactate can be considered a key factor in maternal–fetal communication to promote implantation and the establishment of pregnancy. Indeed, given its small size and high permeability, it may well be the first embryonic signal that the endometrium receives from the embryo.

## Figures and Tables

**Figure 1 biomolecules-15-00100-f001:**
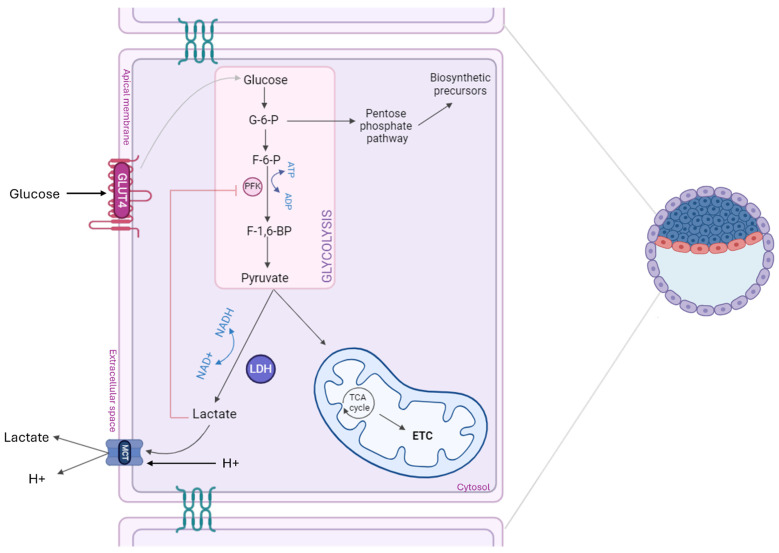
Blastocyst metabolism and lactate production. Despite its high oxidative capacity, approximately 50% of glucose consumed by the blastocyst is metabolized through blastocyst-specific aerobic glycolysis which facilitates the production of biosynthetic precursors through the pentose phosphate pathway (PPP). Pyruvate can be converted to lactate by the action of LDH, which enables the regeneration of cytosolic NAD^+^ to facilitate cyclical continuation of glycolysis, or it can be imported into the mitochondrion for entry into the TCA cycle. Lactate can also allosterically inhibit the action of PFK to regulate glycolytic flux. The export of lactate from the cell is mediated by monocarboxylate transporters (MCTs) with the co-transport of a proton (H^+^). Created with BioRender.com. ETC = electron transport chain; G-6-P = glucose 6 phosphate; F-6-P = fructose 6 phosphate; F-1,6-BP = fructose 1,6 bis phosphate; ATP = adenosine triphosphate; ADP = adenosine diphosphate; GLUT4 = glucose transporter 4; LDH = lactate dehydrogenase; MCT = monocarboxylate transporter; TCA = tricarboxylic acid cycle; PFK = phosphofructokinase.

**Figure 2 biomolecules-15-00100-f002:**
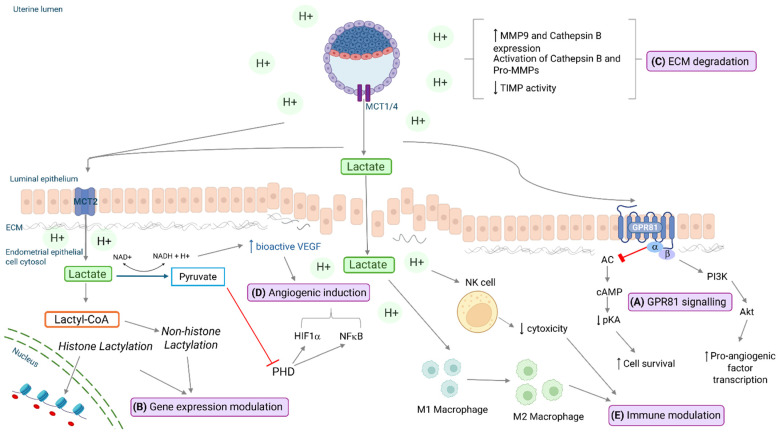
The potential direct and indirect mechanisms of lactate action on cellular functions at the maternal–fetal interface. (**A**) Lactate acts as a ligand for transmembrane GPR81, which impacts downstream cAMP/Akt signaling, PI3K/Akt and other yet to be defined intracellular signaling pathways to promote cell survival, angiogenesis and gene expression modulation in other cell types present at the implantation site. (**B**) Lactate can directly influence gene expression through the lactylation (Kla) of both histone and non-histone proteins. This post-translational modification (PTM) occurs by the action of lactyl-CoA following the cellular uptake of lactate from the extracellular milieu. (**C**) Extracellularly, the acidic microenvironment, due to the co-transport of lactate with protons (H^+^), can affect extracellular matrix (ECM) breakdown through the activation of and increases in MMP activity, and, conversely, by decreasing TIMP activity. (**D**) Lactate uptake by adjacent tissues and cells (e.g., endothelial cells) within the endometrial stroma, through the action of MCT1 and the subsequent intracellular conversion to pyruvate, can indirectly inhibit the action of ADP ribosyl transferases due to the depletion of cytosolic NAD^+^ which increases the availability of bioactive VEGF. This intracellular conversion also inhibits the action of prolyl-hydroxylases (PHD) and results in the stimulation of pro-angiogenic NFκB and HIF1α signaling. (**E**) The low pH/acidic environment at the implantation site also affects the function of numerous immune cells (e.g., NK cells, macrophages) to promote an immune-suppressive phenotype. Further, through a number of mechanisms, lactate produced from the blastocyst also promotes the polarization of M1 macrophages to the M2 phenotype. Created with Biorender.com. PHD = prolyl hydroxylase; HIF1α = hypoxia inducible factor 1 alpha; NFκB = nuclear factor kappa beta; NK = natural killer cell; MMP = matrix metalloprotease; MCT = monocarboxylate transporter; AC = adenylate cyclase; cAMP = cyclic adenosine monophosphate; pKA = protein kinase A; TIMP = tissue inhibitor of metalloprotease; VEGF = vascular endothelial growth factor; PI3K = phosphoinositide 3-kinase; ECM = extracellular matrix.

## Data Availability

No new data were created or analyzed in this study. Data sharing is not applicable to this article.

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
