# Peer review of "Blastocyst-Derived Lactate as a Key Facilitator of Implantation"

_biomolecules, 2025, doi:10.3390/biom15010100_

Round 1

Reviewer 1 Report

Comments and Suggestions for Authors

In this manuscript, the authors review the current understanding of lactate in implantation, summarizing that lactate may facilitate implantation as extracellular ligand and intracellular signaling mediator, as well as through protein lactylation. In addition, acidic pH associated with high lactate concentration may promote trophoblast invasion. While the review is comprehensive and with in depth insight, the manuscript can be improved from the following aspects.

1. Figure 2, it is better to distinguish the intracellular and extracellular functions of lactate and proton in the graph.

2. In the section of lactylation, some recent publications related to embryo development should be cited.

3. Line 289-292, “Once inside these cells, intracellular lactate can be converted to pyruvate, which indirectly stimulates angiogenesis through the activation and recruitment of major pro-angiogenic factors, such as vascular endothelial growth factor (VEGF) and upregulation of pro-angiogenic signaling”. Is this conclusion supported by literatures? Please provide citation. Or this sentence is just a hypothesis. Then please tune down the tone.

4. Grammatical errors need to be corrected, and please pay attention to gene names, with all capital letters or only the first letter capitalized.

Comments on the Quality of English Language

Grammatical errors need to be corrected, and please pay attention to gene names, with all capital letters or only the first letter capitalized.

Author Response

The authors would like to thank the reviewer for taking the time to review this manuscript and for their constructive feedback. Please find below details regarding the revisions we have made to the manuscript based on these comments. Please note that line numbers refer to the tracked changes manuscript, not the clean copy.

Point by point response to comments and suggestions for authors:

In this manuscript, the authors review the current understanding of lactate in implantation, summarizing that lactate may facilitate implantation as extracellular ligand and intracellular signaling mediator, as well as through protein lactylation. In addition, acidic pH associated with high lactate concentration may promote trophoblast invasion. While the review is comprehensive and with in depth insight, the manuscript can be improved from the following aspects.

  1. Figure 2, it is better to distinguish the intracellular and extracellular functions of lactate and proton in the graph.

The authors thank the reviewer for their feedback. We have attempted to make this clearer in the figure legend (lines 205-229) with a more detailed explanation of whether these events occur intracellularly or extracellularly.

  1. In the section of lactylation, some recent publications related to embryo development should be cited.

The authors thank the reviewer for their feedback. Please see our additions to the manuscript on lines 292-303.

  1. Line 289-292, “Once inside these cells, intracellular lactate can be converted to pyruvate, which indirectly stimulates angiogenesis through the activation and recruitment of major pro-angiogenic factors, such as vascular endothelial growth factor (VEGF) and upregulation of pro-angiogenic signaling”. Is this conclusion supported by literatures? Please provide citation. Or this sentence is just a hypothesis. Then please tune down the tone.

The authors apologize for the omission of an appropriate citation following this sentence. We have rectified this and added the necessary citation. Please see our addition on line 364.

  1. Grammatical errors need to be corrected, and please pay attention to gene names, with all capital letters or only the first letter capitalized.

Genes and proteins have been labelled as per the HUGO Gene Nomenclature Committee (HGNC) guidelines for human genes or the International Committee on Standardized Genetic Nomenclature for Mice throughout the manuscript, as summarised below.

  • Human genes symbols contain only uppercase letters and Arabic numerals.
  • Mouse gene symbols begin with an uppercase letter followed by all lowercase letters
  • Protein designations are the same as the gene symbol but are not italicised and are all upper case.

Reviewer 2 Report

Comments and Suggestions for Authors

The prepared work addresses an important and current topic. The work is well organized and described, there is nothing to criticize.

Comments for consideration by the authors:

At the beginning of the work, it would be good to indicate in more detail the gap in the literature on the discussed topic and provide a clear goal of this work.

In my opinion, there is a lack of description of the challenges and further directions of research on the involvement of lactate as a key factor facilitating implantation, which should be demonstrated in future studies, on what further research should focus?

How do the issues described in the article translate into the practical possibility of using this knowledge to facilitate implantation. What is worth investigating/what procedures to perform in the context of implantation disorders?

Author Response

The authors would like to thank the reviewer for taking the time to review this manuscript and for their constructive feedback. Please find below details regarding the revisions we have made to the manuscript based on these comments. Please note that line numbers refer to the tracked changes manuscript, not the clean copy.

Point by point response to comments and suggestions for authors:

The prepared work addresses an important and current topic. The work is well organized and described, there is nothing to criticize. studied

Comments for consideration by the authors:

  1. At the beginning of the work, it would be good to indicate in more detail the gap in the literature on the discussed topic and provide a clear goal of this work.

The authors thank the reviewer for highlighting this omission. We have incorporated this into the updated manuscript on lines 30-35 and 41-48.

  1. In my opinion, there is a lack of description of the challenges and further directions of research on the involvement of lactate as a key factor facilitating implantation, which should be demonstrated in future studies, on what further research should focus? How do the issues described in the article translate into the practical possibility of using this knowledge to facilitate implantation. What is worth investigating/what procedures to perform in the context of implantation disorders?

We are currently working on further research but can’t disclose the details at this stage. However, we agree with the reviewer that this section is a missing piece from our manuscript and have added some detail regarding future directions for research in the conclusion on lines 451-459.

Reviewer 3 Report

Comments and Suggestions for Authors

The review by Gurner and Gardner presents a comprehensive, well-structured and nicely presented overview of recent findings about the role of lactate in for blastocyst implantation.
There remain a few points to be addressed for further improvement of this manuscript.

Major points pertain to the clarity and stringency of a few (bio)chemical aspects that should be improved:

1.      You elaborate on the features of glycolysis in blastocyst that resemble very much those of cancer cells, the well-established “Warburg effect”. While defining it as “aerobic” glycolysis is not really wrong (and I found this in other articles, too) I suggest rephrasing it to atypical aerobic glycolysis. Because common/typical aerobic glycolysis includes conversion of pyruvate to acetyl-Coa and funneling into TCA-cycle/oxphos. However, production of lactate is a hallmark of anaerobic metabolism to replenish NAD+ needed as electron acceptor during fermentation. Indeed, as you describe, atypical aerobic glycolysis is advantageous for adaptation to low oxygen conditions. In my opinion, using the term Warburg effect or atypical aerobic glycolysis provides a clearer semantic separation and facilitates understanding of the unique metabolism in the blastocyst.

2.      Particularly in section 4 you talk about lactate and lactic acid. To me, your differentiation the according to biological function and not chemical features is counterintuitive. You correctly point out that lactic acid has two biological functions it is a) substrate and b) proton donor. Moreover, chemically we have three molecules to consider, not only two: LAH, LA-, H+. I looked up the pka of LAH, which is about 4, meaning at this pH you have exactly half of the molecules as LAH and the other half as LA- and H+. From your text, pH in medium of blastocyst is 6-7, so under physiological conditions you have 100-1000 times more LA- and H+ than LAH. This is something you should mention, because it justifies depicting lactic acid in its dissociated form. Moreover, it explains why MCT proteins are needed, because LAH can easily diffuse across biomembranes, but is under physiological pH in low concentration so that transport facilitators for LA- are required. So chemically lactic acid is LAH and not the dissociated form LA-(lactate) and H+. Therefore, I found your statement “On the other hand, reference to lactic acid (LA), refers to the accumulation of H+ following MCT mediated export of lactate which induces acidification of the extracellular milieu.”  rather hard to grasp. I suggest you explain the features of LAH, LA-, H+ using old-(high) school chemistry. Moreover, in Fig 2 you should depict for MCT2 function not only transport of lactate, but correctly LA- and H+. Then it immediately becomes clear that cytosol is acidified.

Some minor points:
- Line 31: maybe bring key findings from the cited reviews, at least what is relevant to your story. Readers do not like to retrieve the references just to find out if/what may be important to link both sides.
- L 35: please define PAF

-          L50 aside from GPCRs, are there any intracellular receptors (e.g. transcription factors) for lactate – if yes, please mention.

-          L324 and other parts: Apart from covalent (epigenetic) regulation, are there hints towards allosteric regulations by lactate? If not known, maybe something for future research? For instance, I found that lactate allosterically regulates the association of 6PFK enzyme.

-          L136 clarify if 150 pmol glucose consumption is high/low compared to other cells

Author Response

The authors would like to thank the reviewer for taking the time to review this manuscript and for their constructive feedback. Please find below details regarding the revisions we have made to the manuscript based on these comments. Please note that line numbers refer to the tracked changes manuscript, not the clean copy.

Point by point response to comments and suggestions for authors:

Reviewer 3

The review by Gurner and Gardner presents a comprehensive, well-structured and nicely presented overview of recent findings about the role of lactate in for blastocyst implantation.
There remain a few points to be addressed for further improvement of this manuscript.

Major points pertain to the clarity and stringency of a few (bio)chemical aspects that should be improved:

  1. You elaborate on the features of glycolysis in blastocyst that resemble very much those of cancer cells, the well-established “Warburg effect”. While defining it as “aerobic” glycolysis is not really wrong (and I found this in other articles, too) I suggest rephrasing it to atypical aerobic glycolysis. Because common/typical aerobic glycolysis includes conversion of pyruvate to acetyl-Coa and funneling into TCA-cycle/oxphos. However, production of lactate is a hallmark of anaerobic metabolism to replenish NAD+ needed as electron acceptor during fermentation. Indeed, as you describe, atypical aerobic glycolysis is advantageous for adaptation to low oxygen conditions. In my opinion, using the term Warburg effect or atypical aerobic glycolysis provides a clearer semantic separation and facilitates understanding of the unique metabolism in the blastocyst.

We appreciate the reviewer's perspective on the term aerobic glycolysis as used in our manuscript. While we concur that this term should be distinguished from "typical" aerobic glycolysis, we believe that describing it as "atypical" may imply inaccuracy. Consequently, we have revised our manuscript to explain and adopt the term "blastocyst-specific aerobic glycolysis." Please refer to line 93 and the legend for Figure 1(lines 113-125) for these updates.

  1. Particularly in section 4 you talk about lactate and lactic acid. To me, your differentiation the according to biological function and not chemical features is counterintuitive. You correctly point out that lactic acid has two biological functions it is a) substrate and b) proton donor. Moreover, chemically we have three molecules to consider, not only two: LAH, LA-, H+. I looked up the pka of LAH, which is about 4, meaning at this pH you have exactly half of the molecules as LAH and the other half as LA- and H+. From your text, pH in medium of blastocyst is 6-7, so under physiological conditions you have 100-1000 times more LA- and H+ than LAH. This is something you should mention, because it justifies depicting lactic acid in its dissociated form. Moreover, it explains why MCT proteins are needed, because LAH can easily diffuse across biomembranes, but is under physiological pH in low concentration so that transport facilitators for LA- are required. So chemically lactic acid is LAH and not the dissociated form LA-(lactate) and H+. Therefore, I found your statement “On the other hand, reference to lactic acid (LA), refers to the accumulation of H+ following MCT mediated export of lactate which induces acidification of the extracellular milieu.”  rather hard to grasp. I suggest you explain the features of LAH, LA-, H+ using old-(high) school chemistry. Moreover, in Fig 2 you should depict for MCT2 function not only transport of lactate, but correctly LA- and H+. Then it immediately becomes clear that cytosol is acidified.

The authors would like to emphasize the challenge in achieving consensus within the literature on the definition of these three molecules, as different sources use varying terminologies. We are grateful to the reviewer for their suggestion to differentiate lactate and lactic acid based on their chemical properties and for their insightful explanation. The manuscript has been revised (lines 194-238) to incorporate this understanding. We have also updated Figure 2 to include the H+ as transported by the MCT. Thank you for advising us of this omission.

Some minor points:

  • Line 31: maybe bring key findings from the cited reviews, at least what is relevant to your story. Readers do not like to retrieve the references just to find out if/what may be important to link both sides.

Please see lines 41-45

  • L 35: please define PAF

Please see line 50

  • L50 aside from GPCRs, are there any intracellular receptors (e.g. transcription factors) for lactate – if yes, please mention.

To our knowledge, there are no other intracellular receptors that function specifically for lactate. We have however included a small section in our manuscript (lines 269-277) regarding the function of GPR132, a non-specific proton sensing GPCR as a small number of recent papers highlight the role of lactate as one of its numerous ligands.

  • L324 and other parts: Apart from covalent (epigenetic) regulation, are there hints towards allosteric regulations by lactate? If not known, maybe something for future research? For instance, I found that lactate allosterically regulates the association of 6PFK enzyme.

The authors thank the reviewer for bringing this point to their attention. We have now revised our manuscript to include this as a role for lactate in section 3.1 on lines 151-155 and also in Figure 1. Other than regulation of PFK, there seems to be minimal literature regarding this role for lactate in the literature.

  • L136 clarify if 150 pmol glucose consumption is high/low compared to other cells

The authors are unaware of any literature reporting glucose consumption in pmol/cell/hour in other cell types. However, we recognize the importance of a comparative reference to appreciate the magnitude of glucose consumption by the blastocyst. Consequently, we have included a sentence on line 169-174 comparing the glucose uptake of the blastocyst to the cleavage-stage embryo.